# In-situ formatting donor-acceptor polymer with giant dipole moment and ultrafast exciton separation

Chang Cheng [1], Jiaguo Yu [1,2] ✉, Difa Xu[3], Lei Wang[4], Guijie Liang[4], Liuyang Zhang [2] ✉ & Mietek Jaroniec [5] ✉

Donor-acceptor semiconducting polymers present countless opportunities for application in photocatalysis. Previous studies have showcased their advantages through direct bottom-up methods. Unfortunately, these approaches often involve harsh reaction conditions, overlooking the impact of uncontrolled polymerization degrees on photocatalysis. Besides, the mechanism behind the separation of electron-hole pairs (excitons) in donor-acceptor polymers remains elusive. This study presents a post-synthetic method involving the light-induced transformation of the building blocks of hyper-cross-linked polymers from donor-carbon-donor to donor-carbon-acceptor states, resulting in a polymer with a substantial intramolecular dipole moment. Thus, excitons are efficiently separated in the transformed polymer. The utility of this strategy is exemplified by the enhanced photocatalytic hydrogen peroxide synthesis. Encouragingly, our observations reveal the formation of intramolecular charge transfer states using time-resolved techniques, confirming transient exciton behavior involving separation and relaxation. This light-induced method not only guides the development of highly efficient donor-acceptor polymer photocatalysts but also applies to various fields, including organic solar cells, light-emitting diodes, and sensors.

Photocatalysis, a sunlight-driven process, has witnessed substantial progress in the field of sustainable energy, materials, and chemistry. However, its performance is limited by rapid electron-hole recombination. Until now, photocatalysts, either inorganic semiconductors or organic polymers, have been extensively studied. Specifically, donor-acceptor (D-A) polymers with an alternating array of electron-rich and electron-deficient moieties have gained particular attention[1–4]. The light absorption and band gaps of the polymers can be easily tuned by modifying their molecular structures[5–7].

Although photocatalysts with D-A structure have been continuously studied, most works use tedious and expensive methods to synthesize polymers with or without D-A structure and associate their activities with the spectral absorption or band structure, rather than investigating the underlying photophysical processes[8–11]. Of note, even the same polymer synthesized in different batches displays slightly different band structures due to different polymerization degrees. Therefore, it is desirable to construct D-A structures through a post-synthetic transformation method that directly modifies the as-prepared polymer rather than the monomers, thereby avoiding changes in the polymer backbone (degree of polymerization). Considering the industrial perspective, there is a need to develop a simple and cheap strategy to prepare polymer materials with robust

[1]State Key Laboratory of Advanced Technology for Materials Synthesis and Processing, Wuhan University of Technology, 430070 Wuhan, P. R. China. [2]Laboratory of Solar Fuel, Faculty of Materials Science and Chemistry, China University of Geosciences, 388 Lumo Road, Wuhan 430074, P. R. China. [3]Hunan Key Laboratory of Applied Environmental Photocatalysis, Changsha University, 98 Hongshan Road, Changsha 410022, P.R. China. [4]Hubei Key Laboratory of Low Dimensional Optoelectronic Material and Devices, Hubei University of Arts and Science, Xiangyang 441053, P. R. China. [5]Department of Chemistry and Biochemistry, Kent State University, Kent, OH 44242, USA. ✉e-mail: yujiaguo93@cug.edu.cn; zhangliuyang@cug.edu.cn; jaroniec@kent.edu

photocatalytic activity. Hyper-cross-linked polymers (HCPs), an emerging type of porous organic polymers (POPs), have emerged as promising photocatalysts due to their broad light absorption, facile synthesis, and low cost. They are usually synthesized via *Friedel-Crafts* or *Scholl* reactions[12–14]. Recent progress has allowed less hazardous cross-linker and more facile methods to be used. For example, Tan and co-workers proposed an effective strategy for knitting aromatic units by employing formaldehyde dimethyl acetal (FDA) as an external cross-linker to simplify the synthesis of HCPs[15]. Thus, a combination of this simplified production of HCP materials with post-synthetic transformation presents a platform for scale-up exploration and application.

Herein, dibenzothiophene (DBT) was selected as an aromatic monomer owing to its remarkable optical and electronic properties[16–19]. Upon linkage of DBT, the electrons around the polymer skeleton are delocalized due to the hyperconjugation effect ($\sigma \to \pi^*$), thus leading to a considerable absorption improvement in the visible region. Interestingly, we discovered a one-pot photoredox-activated method for the transformation from donor-donor to donor-acceptor polymer. Precisely, pristine knitted DBT (KDBT) was converted into KDBT-A with a significantly enhanced $H_2O_2$ production rate. As underpinned by XPS and FT-IR spectra, a portion of electron-rich DBT units was in situ photocatalytically oxidized to dibenzothiophene-*S,S*-dioxide (DBTSO), which resulted in changes of optoelectronic properties and assured better photocatalytic performance. Besides, DFT calculations demonstrated that DBTSO is a strong electron-

withdrawing monomer. Thus, once DBT-carbon-DBT (donor-C-donor) is oxidized to DBT-carbon-DBTSO (donor-C-acceptor), the dipole moment increases by ~10 times. The strong separation of excitons induced by high dipole moment has been systematically verified by both steady-state and transient techniques. This study confirms the feasibility of the proposed transformation strategy, which leads to the profound correlation between photocatalytic activity and the optoelectronic nature of semiconducting polymers.

## Results

### Synthesis and characterization of KDBT

The original KDBT polymer was synthesized via a *Friedel-Crafts* reaction (Supplementary Fig. 1). Figure 1a shows the connection (knitting) of dibenzothiophene by the FDA. Upon continuous ultrasonication, KDBT was gradually grown into a tubular structure (Fig. 1b−f). Powder X-ray diffraction (PXRD, Supplementary Fig. 2) measurements indicate its amorphous nature[20]. Fourier transform infrared (FTIR, Supplementary Fig. 3) spectrum of KDBT exhibits strong bands at 1485 and 2920 cm$^{-1}$, which can be assigned to aromatic ring skeleton vibration and C-H stretching vibration of alkyl groups[21]. The $^{13}$C cross-polarization magic-angle-spinning nuclear magnetic resonance (CP/MAS NMR, Supplementary Fig. 4) spectra of KDBT exhibit intense resonance peaks at ~138 and ~124 ppm, which is attributed to the substituted and unsubstituted aromatic C, respectively. The peak at ~39 ppm is interpreted as the signal of methylene linker C. These results indicate that the FDA successfully knitted DBT units. The cross-

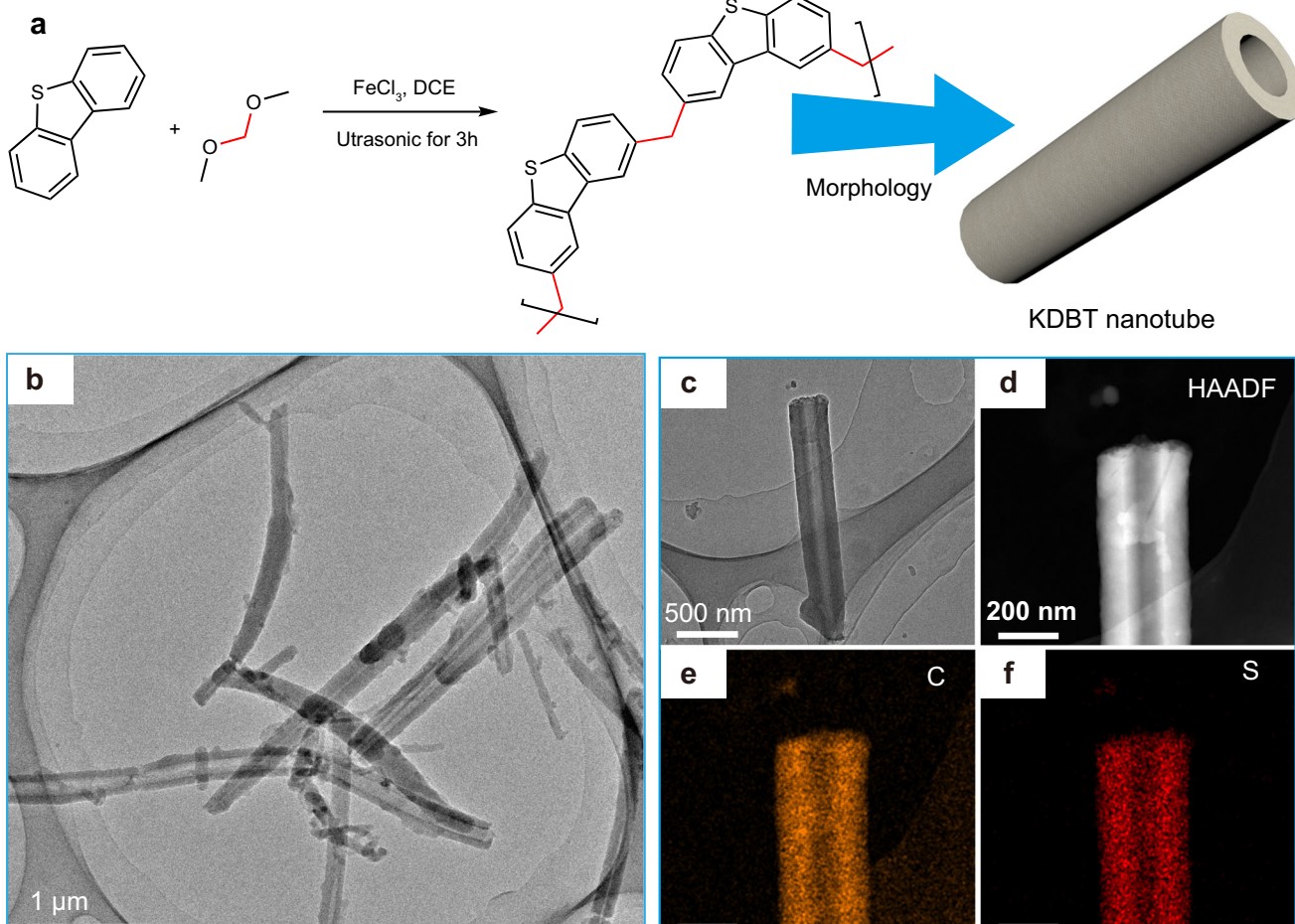

**Fig. 1 | Synthesis and morphology of KDBT polymer. a** FeCl$_3$-catalyzed *Friedel-Crafts* polycondensation route for KDBT. **b, c** Transmission electron microscopy (TEM) images of KDBT. HAADF image (**d**) and the corresponding elemental carbon (**e**) and sulfur (**f**) maps of the KDBT nanotube.

linked structure endows KDBT with great stability (Supplementary Fig. 5) and porosity (Supplementary Fig. 6)[22].

After characterizing the molecular structure of KDBT, we further investigated its band structure. As shown in Supplementary Fig. 7a, KDBT exhibits strong absorption in the visible range and tails off toward 800 nm. Benefiting from the hyperconjugation effect ($\sigma_{C-H} \rightarrow \pi^*$), the electrons around the polymer backbone are partially delocalized, manifesting a medium bandgap of KDBT (2.75 eV, Supplementary Fig. 7b). The band positions of the polymer were determined by various methods. Tested by ultraviolet photoelectron spectroscopy (UPS, Supplementary Fig. 8a), the cutoff energy ($E_{cutoff}$) and highest occupied state (HOS) of the polymer are located at 19.53 and 4.60 eV, respectively. Thus, the ionization potential of KDBT (equal to the highest occupied molecular orbital (HOMO) level) is calculated to be −6.34 eV vs. vacuum, corresponding to +1.84 V vs. normal hydrogen electrode (NHE)[23–25]. Accordingly, its lowest unoccupied molecular orbital (LUMO) level is determined to be −0.91 V vs. NHE. A similar band structure of KDBT was also obtained from Mott-Schottky (M-S, Supplementary Fig. 8b) plots and cyclic voltammetry (CV, Supplementary Fig. 9) curves, and these results are summarized in Supplementary Table 1.

## Photocatalytic Performance

Given the ideal stability, porosity, and suitable band structure of KDBT, the potential of this polymer as a photocatalyst for hydrogen peroxide production is evaluated. As shown in Fig. 2a, no $H_2O_2$ was generated in the dark. Upon light irradiation in air for 1 h, the $H_2O_2$ concentration reached ~870 μM. Interestingly, it further increased sharply by approximately 50% in the second cycle during the cycling test (sampling every 15 min and 60 min per cycle), and became stable over the next 8 cycles (Fig. 2b), which differs from the reported results (Supplementary Table 2)[26–36]. The reason for this enhancement is the structural transformation of the D-D to the D-A structure as shown in Fig. 2c. This transformed form of the polymer is denoted as KDBT-A, and the transformation process is discussed later. In general, photocatalytic performance is highly dependent on the band structure (redox ability) of photocatalysts. Hence, we first compared the band structures of KDBT and KDBT-A. As shown in Supplementary Fig. 6, two absorption spectra almost overlapped, indicating that the optical band gap of the sample did not change after the reaction. Moreover, the CV curve (Supplementary Fig. 9), UPS, and M-S plots (Supplementary Fig. 10) suggest that the LUMO and HOMO level of KDBT-A is +1.87 and −0.88 V vs. NHE, respectively. From a thermodynamic point of view, the band positions (driving force for $H_2O_2$ production, Fig. 2d) of KDBT and KDBT-A are quite close. However, electron paramagnetic resonance (EPR) analysis yielded different results (Fig. 2e and Supplementary Fig. 11). Specifically, 5,5-dimethyl-l-pyrroline N-oxide (DMPO) was used to capture the superoxide radical (•$O_2^-$) and hydroxyl radical (•OH). KDBT exhibits a clear DMPO-•$O_2^-$ signal in the air because its LUMO level is more negative than the electrode potential of $O_2$/•$O_2^-$ (−0.33 V at pH = 7). Interestingly, although the HOMO level of KBDT was higher than the standard potential of $H_2O$/•OH ( + 2.34 V at pH = 7), a DMPO-•OH signal was still detected in the air. When tested in argon (Ar), the DMPO-•OH signal disappeared, implying that the •OH stems from $O_2$ reduction rather than $H_2O$ oxidation. As for KDBT-A, even though KDBT-A exhibits a lower LUMO position (reducing power) than pristine KDBT, its EPR signals of •$O_2^-$ and •OH are markedly enhanced. It is inferred that: (1) Band structure is not the dominant factor affecting $H_2O_2$ production efficiency. (2) The photoinduced electron transfer (PET) process is more effective for KDBT-A than KDBT.

## In situ evolution of KDBT to KDBT-A

To support this hypothesis, we carefully explored the molecular structures of KDBT and KDBT-A. As shown in Supplementary Fig. 2,

KDBT-A presents a hump at the same 2θ degree as KDBT. Comparing the FTIR spectra of KDBT, KDBT-A, and their respective monomers confirms that KDBT-A retains the molecular structure of KDBT even after prolonged irradiation, a fact further supported by the $^{13}$C CP/MAS NMR spectra of the polymer after each cycle. More importantly, an absorption peak at ~1150 cm$^{-1}$ emerges on the spectrum of KDBT-A, which is ascribed to the characteristic signal of the sulfonyl group (O = S = O)[37]. It shows that a part of dibenzothiophene units is oxidized to dibenzothiophene-*S,S*-dioxide during the photocatalytic process. XPS analyses provide solid evidence for this transformation. The survey spectra (Supplementary Fig. 12) revealed the presence of C and S elements in KDBT, and the S/C atomic ratio (19.7%) is close to the theoretical ratio of 20.5%. Notably, a weak O 1$s$ peak at ~532.9 eV is observed and attributed to the terminal alkoxy group of FDA. The S/C atomic ratio in KDBT-A (20.0%) is also close to the theoretical ratio, again confirming the high stability of the polymer skeleton. However, the O 1$s$ peak is significantly amplified in KDBT-A, suggesting the incorporation of O atoms into the polymer chains. In the high-resolution XPS spectrum of C 1$s$ (Fig. 2f), three peaks of KDBT are located at 284.8, 286.5, and 289.0 eV, respectively. The first peak belongs to adventitious carbon for calibration. The latter two peaks originate from C-S bonds of dibenzothiophene moiety and $sp^3$ carbon linkages. The S 2$p$ spectrum of KDBT (Fig. 2g) is divided into two peaks at 163.9 (S 2$p_{3/2}$) and 165.0 eV (S 2$p_{1/2}$), which are attributed to C-S-C of dibenzothiophene unit. For comparison, the C 1$s$ spectrum of KDBT-A is similar to that of KDBT. Interestingly, a new set of S 2$p$ signals appears at 167.9 and 169.0 eV (see Fig. 2f), which can be assigned to the sulfonyl group (Fig. 2c)[38].

## Photocatalytic mechanism

It is crucial to unravel the mechanism of structural transformation of the polymer photocatalyst. We conducted in situ diffuse-reflectance infrared Fourier transform spectroscopy (DRIFTS) to monitor the irradiation process. To ensure an accurate interpretation of the in situ DRIFTS spectra, we conducted FTIR spectra measurements on both $H_2O_2$ and $H_2O$ at first. As shown in Supplementary Fig. 13a, they exhibit similar IR absorption beyond 3000 cm$^{-1}$. Hence, we directed our focus towards the characteristic peak of $H_2O_2$, specifically identified at ~2850 and 1380 cm$^{-1}$. As shown in Supplementary Figs. 13b, c, the emergence of signals around ~2850 and 1380 cm$^{-1}$ corresponds to the gradual formation of $H_2O_2$. Interestingly, the signal from the sulfonyl group also emerged and its intensity gradually increased with continuous irradiation, providing compelling evidence of DBT oxidation. The in situ oxidation of KDBT to KDBT-A was also tracked by XPS (Fig. 3a). The oxidation degree can be roughly quantified by the percentage increase of the newly emerged S 2$p$ peak, which exhibited a gradual rise during the first hour (0−6%−9%−15%−26%), and this percentage remained almost unchanged in the subsequent cycle tests (Supplementary Fig. 14). Since the oxidation of DBT occurs in the photocatalytic experiment, the reaction route of $H_2O_2$ photosynthesis was investigated through a series of control experiments. In pure water, KDBT-A exhibits a high $H_2O_2$ production rate of 1320 μM h$^{-1}$, about 1.5 times that of KDBT (841 μM h$^{-1}$). As shown in Fig. 3b, neither KDBT nor KDBT-A is active in the presence of AgNO$_3$ (electron scavenger, Ag$^+$). With the introduction of methanol (MA) as a hole scavenger, the production rate increases to 1881 μM h$^{-1}$ for KDBT-A and 1152 μM h$^{-1}$ for pristine KBDT. Of note, the molecular structure of KDBT remains intact even after prolonged irradiation (2 h) in MA aqueous solution (Supplementary Fig. 15). The introduction of isopropanol (IPA) as a •OH scavenger does not affect photocatalytic $H_2O_2$ production (KDBT: 801 μM h$^{-1}$; KDBT-A: 1280 μM h$^{-1}$). In contrast, only trace amounts of $H_2O_2$ are detected on KDBT and KDBT-A using benzoquinone (BQ) as a •$O_2^-$ scavenger. These results indicate that •OH does not promote the formation of $H_2O_2$, whereas •$O_2^-$ is the primary reactive oxygen species (ROS). Oxidation half-reactions were explored under Ar conditions.

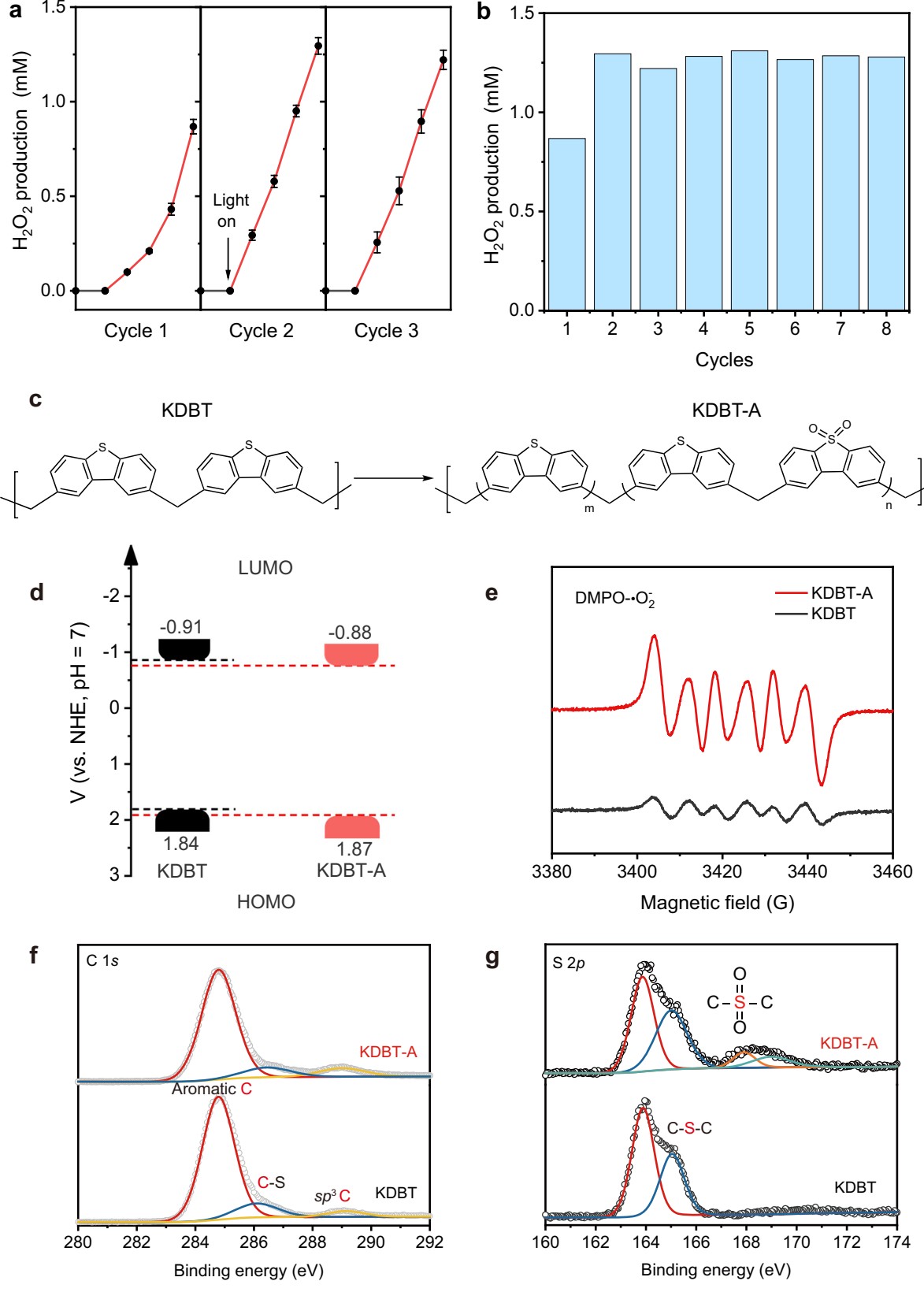

**Fig. 2 | Photocatalytic performance, driving force, and transformation of the polymers. a** Time course of $H_2O_2$ production over the polymer. The base point was set when the system reached absorption-desorption equilibrium in the dark. The error bars (mean ± standard deviation) were obtained from three independent photocatalytic experiments. **b** 8 long-time cycle tests. **c** The possible structure transformation from KDBT to KDBT-A. **d** Band structure diagrams of KDBT and KDBT-A. **e** Light-irradiated EPR spectra of DMPO-•$O_2^-$ adducts. Experimental conditions: 2 mg of samples, 10 mL of anhydrous $CH_3CN$, irradiated by LED ($\lambda = 365$ nm) for 5 min. High-resolution X-ray photoelectron spectra (XPS) of C 1$s$ (**f**) and S 2$p$ (**g**) in KDBT and KDBT-A.

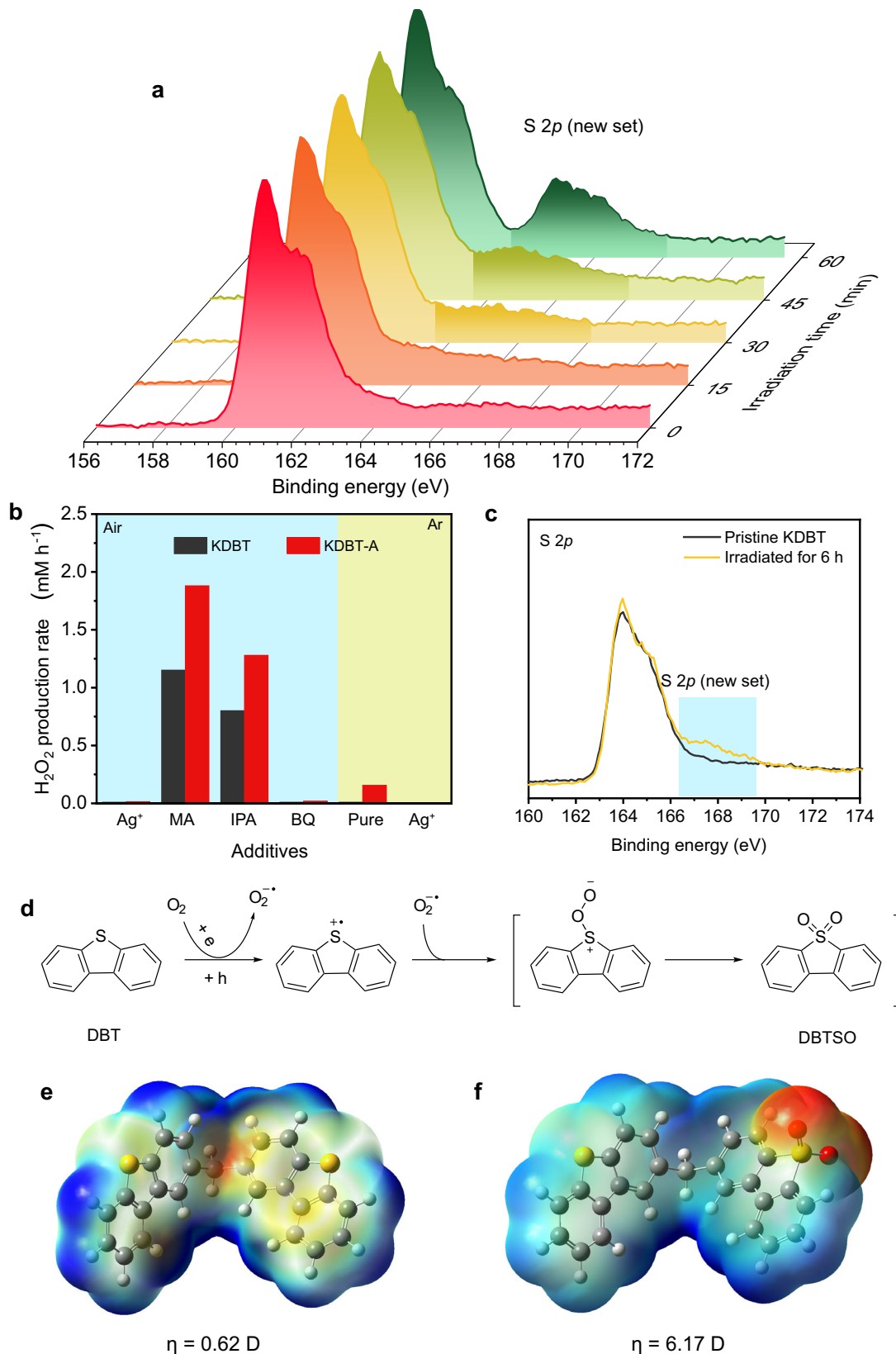

**Fig. 3 | In situ transformation process of KDBT to KDBT-A and the corresponding mechanism. a** Evolution of S 2p XPS spectrum of the polymer during photocatalysis. A new and weaker signal gradually emerges on the spectrum, indicating that the polymer is progressively oxidized. **b** Blue background: Photocatalytic performance for KDBT and KDBT-A in AgNO₃ (0.01 M), methanol (10 v.%), isopropanol (10 v.%), or benzoquinone (0.01 M) aqueous solution under air condition. Yellow background: Photocatalytic performance for KDBT and KDBT-A in pure water or AgNO₃ (0.01 M) aqueous solution under Ar condition. **c** S 2p XPS spectra of KDBT before and after irradiation in Ar atmosphere for 6 h. **d** Oxidation pathway of DBT in photocatalysis. **e, f** Theoretical calculations performed to investigate the electrostatic potential distribution and dipole moments of KDBT and KDBT-A at the B3LYP/631 G(d,p) level.

After 1 h of irradiation, KDBT-A exhibits small $H_2O_2$ production (156 μM h⁻¹) in pure water; whereas KDBT is completely inactive. Interestingly, KDBT is slightly oxidized after further irradiation for 6 h (Fig. 3c). Introduction of $AgNO_3$ into the anaerobic system leads to $O_2$ production in both samples, which is testified by isotope analysis (Supplementary Figs. 16 and 17). Based on these data, we propose a mechanism for structural transformation. In the early stage of irradiation, a part of DBT units traps photogenerated holes to form the radical cations, DBT⁺• (Supplementary Fig. 1b). Then, a derivative of photogenerated $O_2$, i.e. •$O_2^-$ can quickly attack DBT⁺• at the S position to form DBTSO units (Fig. 3d)[39,40].

### Photophysical process validation

Typically, DBT is an electron-rich monomer, while DBTSO is an electron-deficient monomer. In the original KDBT polymer, its building block, DBT-carbon-DBT, has a small dipole moment of 0.62 D due to its symmetrical structure (Fig. 3e)[41]. After photocatalysis, the building block is partly oxidized to DBT-carbon-DBTSO. Owing to the different electron affinities between donor and acceptor, it exhibits a significant

dipole moment of 6.17 D, (Fig. 3f)[42]. Namely, the polymer molecule transforms from a pure donor-donor structure to a donor-donor/donor-acceptor hybrid structure. This molecular transformation results in changes in the photophysics properties of KDBT-A.

Steady-state phosphorescence and fluorescence (Fig. 4a, b) spectra were measured to elucidate the photogenerated charge transfer behavior within KDBT and KDBT-A. First, no phosphorescence is detected on two samples, indicating that no triplet excited state is involved in the photoexcitation process. Under oxygen saturation conditions, two fluorescence peaks are detected on the spectrum of KDBT. The fluorescence peak F1 at 528 nm is attributed to the recombination of photogenerated electron-hole pairs (bound state excitons, e-h). The peak at 428 nm (F2) reflects the emission photon energy of 2.90 eV, which exhibits an anti-Stokes shift of 150 meV relative to the optical band gap of KDBT. Typically, the anti-Stokes shift arises from two pathways (Supplementary Fig. 18): (1) triplet-triplet annihilation upconversion (TTA-UC), (2) higher-level carrier recombination (higher singlet excited state, $S_n$)[43,44]. As mentioned above, phosphorescence tests have already confirmed the absence of a triplet

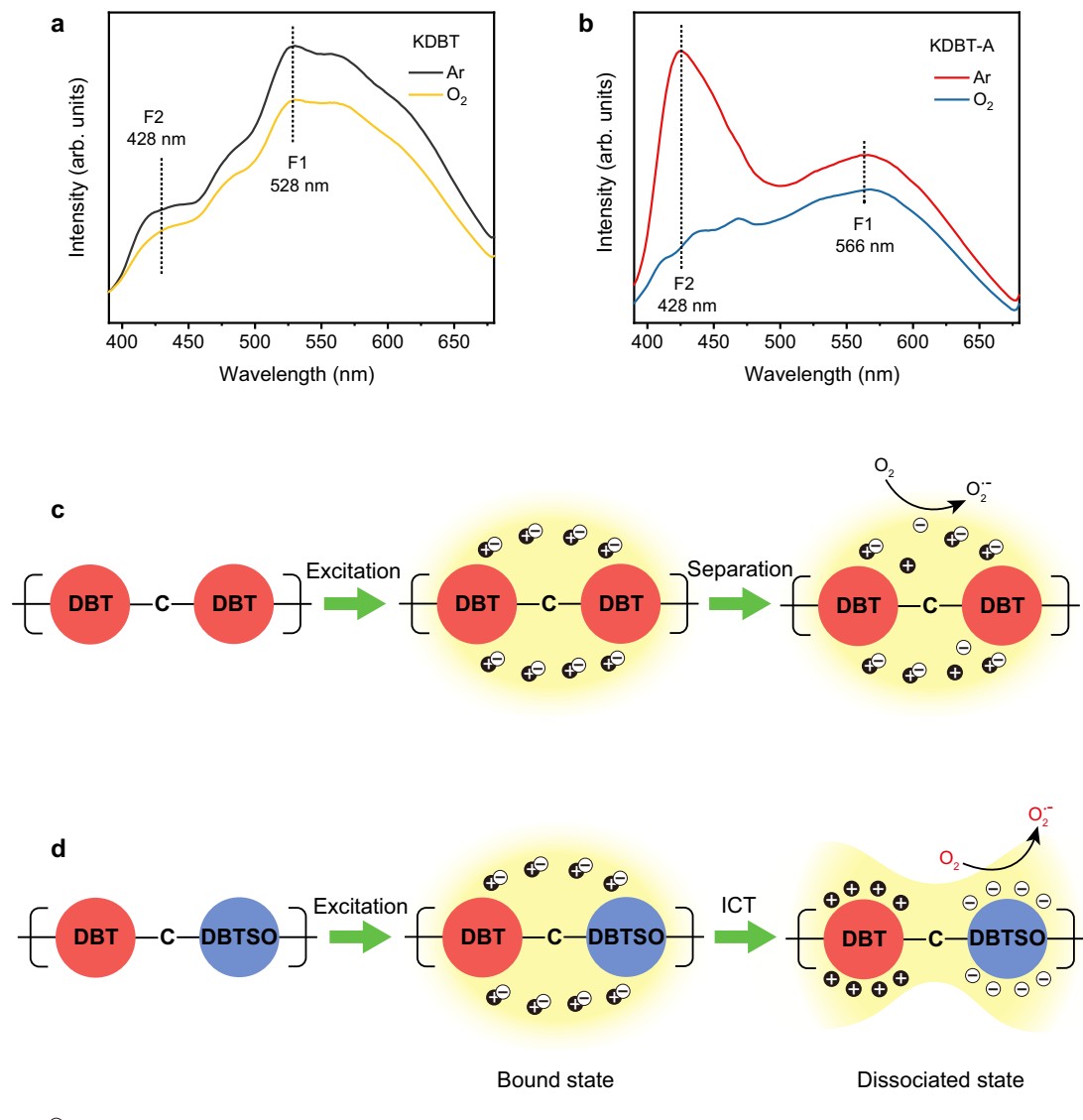

**Fig. 4 | Macroscopic insights into the photophysical processes. a, b** Steady-state fluorescence spectra of KDBT and KDBT-A under air and Ar conditions, where F1 and F2 denote the observed fluorescence peaks. **c, d** Schematic representation of photocatalytic processes (excitation, electron transfer) in KDBT and KDBT-A polymers.

excited state; hence the first pathway is ruled out. In the second pathway, higher energy level charge carriers correspond to the dissociated excitons, that is, dissociated photogenerated electrons and holes ($e^-$ $h^+$). Therefore, the weak F2 peak suggests a low degree of exciton separation within KDBT (Fig. 4c). It is noteworthy that the fluorescence intensity of KDBT slightly increases under Ar conditions because no photogenerated electrons are trapped by $O_2$. Regarding KDBT-A, its spectral signature in the $O_2$ atmosphere is similar to that of KDBT. However, under Ar conditions, the F2 peak ($\lambda = 428$ nm) becomes significantly stronger than F1 ($\lambda = 566$ nm), demonstrating that more excitons are dissociated in KDBT-A than in KDBT. This phenomenon can be attributed to the giant dipole moment of DBT-C-DBTSO units, which can trigger intramolecular charge transfer (ICT) upon excitation. As a result, photogenerated electrons and holes are concentrated at acceptor and donor moieties, respectively. In this

case, excitons are efficiently dissociated, and more photogenerated electrons are trapped by $O_2$ (Fig. 4d).

Kelvin probe force microscopy (KPFM) can visualize charge transfer and separation[45]. Figure 5a–f displays the topography of KDBT and KDBT-A and the corresponding surface potential images. Before irradiation, the maximum contact potential differences (CPDs) are measured to be -180 mV for KDBT and -269 mV for KDBT-A. Upon light ($\lambda = 365$ nm) irradiation, no distinct difference in CPDs (equivalent to SPV) is observed on KDBT (Fig. 5g), while the CPDs of KDBT-A increase by -136 mV (Fig. 5h), indicating that more excitons are separated within KDBT-A compared to KDBT. Photoelectrochemical measurements (Supplementary Fig. 19a, b) also confirm the facilitated charge separation.

To gain deeper insight into the photophysical process, we applied time-resolved techniques to investigate the transient dynamics of the

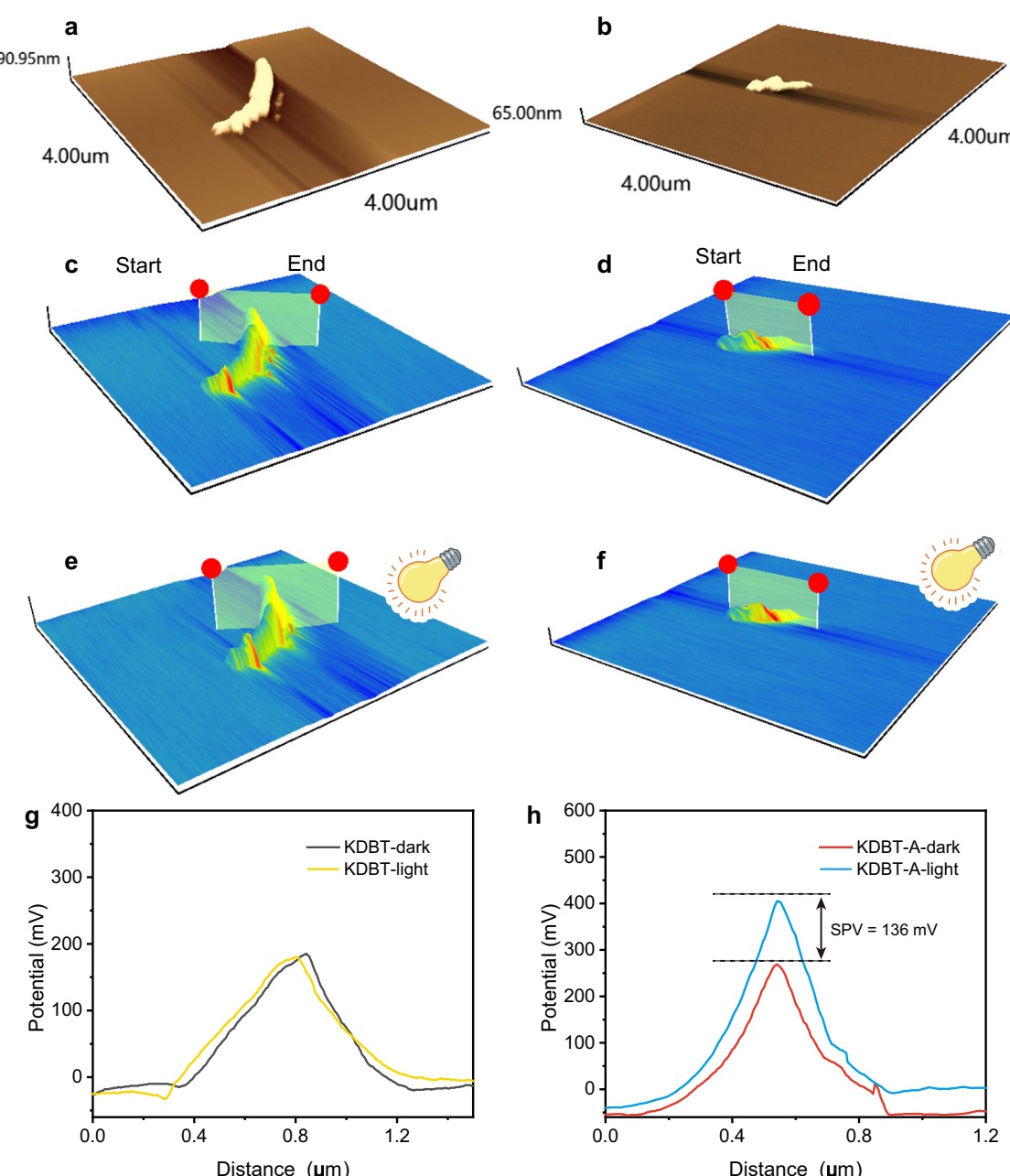

**Fig. 5 | Visualization of charge separation. a, b** Atomic force microscopic images of KDBT and KDBT-A. **c–f** Corresponding surface potential distributions of KDBT and KDBT-A. Line-scan surface potential across KDBT (**g**) and KDBT-A (**h**) nanotube. Surface photovoltage (SPV) could be obtained as follows: $SPV = CPD_{light} - CPD_{dark}$.

excited state. First, the exciton relaxation process was revealed using time-resolved fluorescence (TRFL) spectroscopy. The calculated fluorescence lifetime of F1 ($\tau_f$, Supplementary Fig. 20a) was 0.63 ns for KDBT and 0.91 ns for KDBT-A. The short fluorescence lifetime of KDBT reflects rapid exciton recombination, usually caused by Coulombic attraction between photogenerated electrons and holes (Supplementary Fig. 20b). Whereas, in the KDBT-A molecule, the magnificent dipole moment directing from DBTSO to DBT counteracts the Coulombic force, thereby reducing the exciton recombination rate and prolonging the lifetime of F1. As shown in Supplementary Fig. 21a, F2 peak decays of two samples are well fitted by bi-exponential function, demonstrating that two pathways dominate the attenuation of separated excitons ($S_n$, Supplementary Fig. 21b). Wherein the short lifetime ($\tau_s$) generally reflects nonradiative relaxation ($S_n \rightarrow S_1$), and the long lifetime ($\tau_l$) is assigned to the direct recombination of separated charge carriers ($S_n \rightarrow S_0$). $\tau_s$ and $\tau_l$ of KDBT are estimated to be 0.38 and

3.62 ns, which account for 89.88% and 10.12%, respectively. The fitting values of $\tau_s$ and $\tau_l$ for KDBT-A are 0.40 and 4.50 ns, corresponding to the proportion of 69.06% and 30.94%. On the one hand, KDBT-A has a longer lifetime, implying that its photogenerated charge carriers are more prone to being trapped (by $O_2$). On the other hand, the proportion of nonradiative relaxation within KDBT-A is lower than that within KDBT, thus validating the efficient charge separation on KDBT-A.

Femtosecond transient absorption (fs-TA) spectroscopy was employed to comprehend the unresolved gap in the ultrafast exciton separation process[46]. Based on steady-state photoluminescence spectroscopy, the exciton splitting of polymers ($S_1 \rightarrow S_n$) requires low energy intake, corresponding to light absorption in the near-infrared (NIR) region. Therefore, the NIR probe can offer microscopic views of the separation behavior of excitons. As depicted in Fig. 6a–f and Supplementary Fig. 22, KDBT and KDBT-A exhibit broad absorption

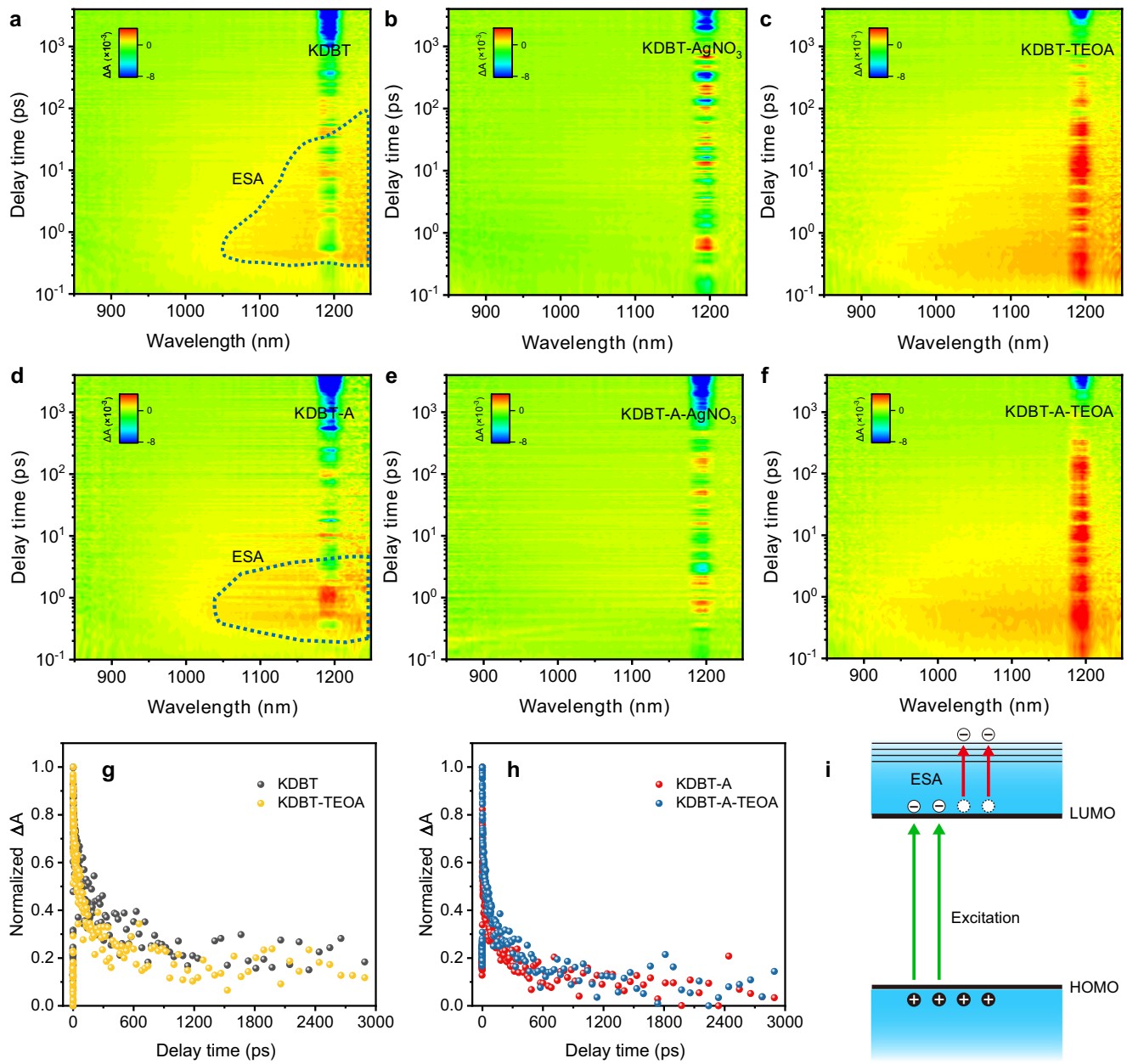

**Fig. 6 | Validation of ultrafast excitonic behavior. a–c** Two-dimensional (2D) TA spectra mapping of KDBT with or without scavengers. **d–f** 2D TA spectra mapping of KDBT-A with or without scavengers. All data were acquired under excitation of 400 nm and optical power of 260 µW cm$^{-2}$. **g, h** Extracted decay kinetics of KDBT and KDBT-A with or without TEOA at $\lambda = 1150$ nm. **i** Schematic illustration of the ESA signal formation process.

signals after 950 nm, which belong to the excited-state absorption (ESA, Supplementary Fig. 18b), reflecting their transition from the bound state (e-h) to dissociated state (e⁻ h⁺). The cluttered signals at ~1200 nm are inevitable overtones of the pump pulse ($\lambda = 400$ nm). Typically, the assignment of TA spectral features is complex because they involve the total energy variation of electrons and holes. To investigate the carrier contributions to ESA, we performed scavenger-assisted fs-TA spectra using $AgNO_3$ or triethanolamine (TEOA) to selectively remove photogenerated electrons or holes, respectively. The results show that KDBT and KDBT-A ESA signals are effectively quenched in the presence of $AgNO_3$ (Supplementary Fig. 23), while the signals remain with TEOA scavenger (Supplementary Fig. 24). Figure 6g, h compares ESA decays (to eliminate the effect of overtones, we extracted decay curves at $\lambda = 1150$ nm) of the materials with or without TEOA. The decay traces of bare polymers overlapped with those in the presence of TEOA. These spectral features suggest that the ESA signal is mainly attributed to the electron transition to a higher energy level (Fig. 6i) with a trivial contribution from holes.

Analysis of the scavenger-assisted ESA decay indicates that the intensity of the ESA signal is highly dependent on the population of LUMO electrons. As shown in Fig. 7, the decay plots of KDBT and KDBT-A fit a triple-exponential function, indicating that the reduction of LUMO electrons in the polymers proceeds via three pathways. First, the most common pathway, namely, the recombination of excitons, has been characterized by TRFL spectroscopy ($\tau_3 = \tau_f$). Second, electron diffusion (delocalization) over molecular chains occurred on an ultrafast timescale (several picoseconds), which is consistent with the

shortest lifetime ($\tau_1$). Third, the last remaining parameter ($\tau_2$) is interpreted as the lifetime of electron transition from HOMO to upper energy level. As a result, $\tau_2$ is fitted to be 101 ps for KDBT and 85 ps for KDBT-A, and the electron transition rate can be calculated ($k_T$) to be $9.9 \times 10^9 \, s^{-1}$ and $11.8 \times 10^9 \, s^{-1}$, respectively. Further, the proportion of electron transition pathway in KDBT-A is ~24%, which is higher than that in KDBT (10%). Consequently, the enhanced electron transition rate and proportion in KDBT-A confirm the effectiveness of the ICT state (or giant intramolecular dipole moment) in facilitating exciton separation.

## Discussion

In summary, this study presents a paradigm for investigating the formation and efficacious charge separation of light-triggered D-A structures (or electronic push-pull interactions). Experiments coupled with DFT calculations verify that a part of dibenzothiophene can be in situ oxidized to the electron-withdrawing dibenzothiophene-*S,S*-dioxide units (KDBT-A) during photocatalysis, which results in increasing the molecular dipole moment and creating an intramolecular charge transfer (ICT) state. Also, apart from steady-state approaches (PL, KPFM, and photoelectrochemical measurements), time-resolved techniques (TRFL and fs-TAS) provide insights into the effectiveness of the ICT state in suppressing exciton recombination. As a result, the photocatalytic activity is enhanced. Although the protocol is only used to synthesize one type of D-A photocatalysts, we anticipate that this conceptual approach is useful for the development of other D-A polymer systems.

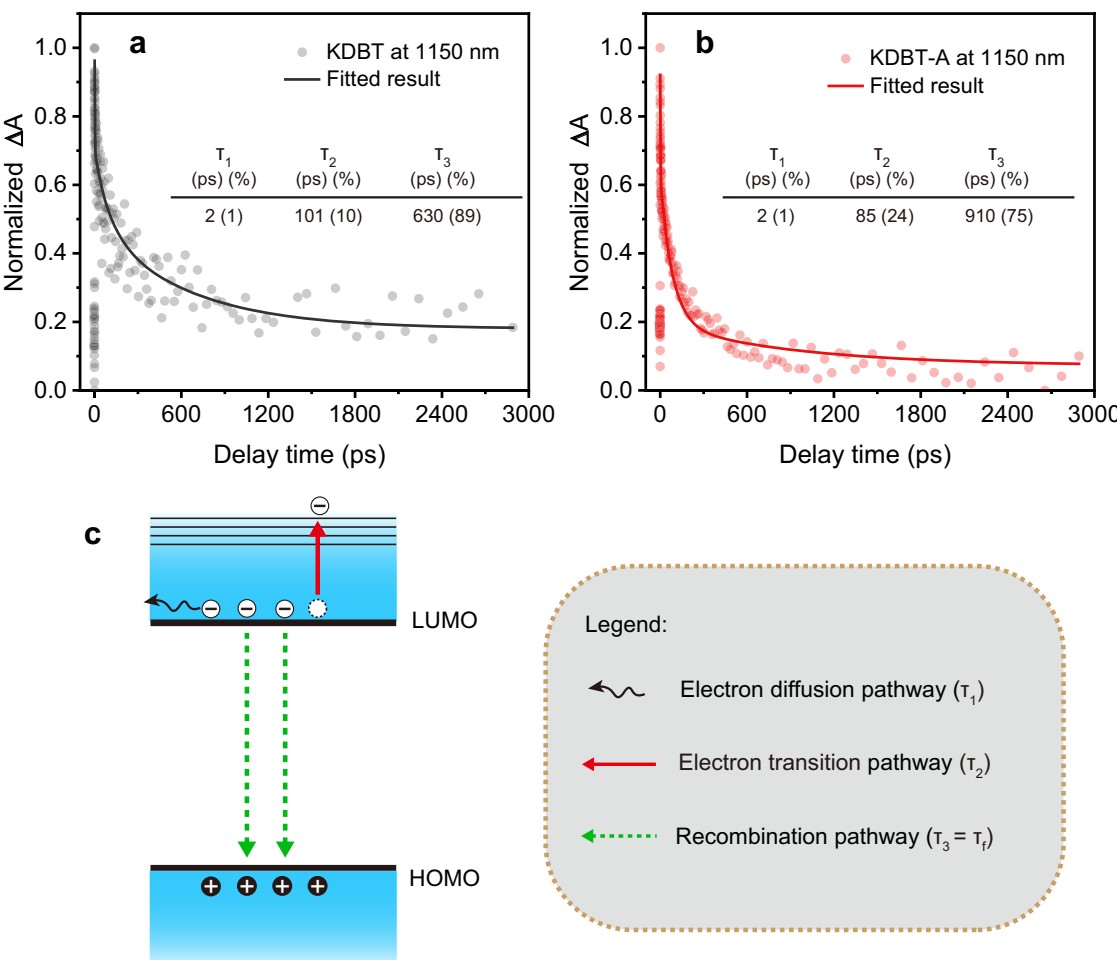

**Fig. 7 | Microscopic insights into the photophysical processes.** Normalized decay kinetic plots for KDBT (**a**) and KDBT-A (**b**) at 1150 nm. **c** Schematic illustration for the decay pathways of photogenerated electrons.

## Methods

### Materials

DBT (≥ 99%), FDA (98%), anhydrous FeCl$_3$ (analytical grade), KI (analytical grade), C$_8$H$_5$KO$_4$ (99.8%), DMPO (97%), AgNO$_3$ (analytical grade), La$_2$O$_3$ (99.99%), BQ (99 %), and ferrocene (99 %) were obtained from Aladdin Reagent Co. Ltd. 1,2-dichloroethane, HCl, H$_2$O$_2$, methanol, acetonitrile, TEOA were purchased from National Medicines Corporation Ltd. of China. All of the above reagents were of analytical grade and used in their original state.

### Synthesis of knitted dibenzothiophene (KDBT)

Polymer KDBT was synthesized via *Friedel-Crafts* reaction[15]. A series of control experiments were performed to investigate the effect of synthetic conditions on the morphology of KDBT polymer (Supplementary Table 3). In the optimized procedure, dibenzothiophene (184 mg, 1 mmol), and formaldehyde dimethyl acetal (FDA, 2 mmol) were dissolved in 40 mL of 1,2-dichloroethane (DCE). After 5 min of ultrasonic treatment, anhydrous FeCl$_3$ (324 mg, 2 mmol) was added. The mixture was ultrasonicated (480 W) for another 3 h, and the temperature was set to 35 °C. After that, the solid was collected by filtration and washed once with HCl-H$_2$O (v/v = 2:1) and twice with anhydrous methanol to remove Fe residues and oligomers. After filtration and drying, insoluble yellow powders were obtained at an 84% yield, amounting to 164 mg.

### Photocatalytic H$_2$O$_2$ production

The photocatalytic experiments were conducted in a 100 mL breaker using a 300 W Xenon arc lamp (149 mW cm$^{-2}$) as the light source. In a typical photocatalytic reaction, 5 mg of photocatalyst was dispersed in 30 mL of ultrapure H$_2$O. The system was directly irradiated under air conditions, and the distance between the light source and the flask was set as 10 cm. The generation of H$_2$O$_2$ was investigated by an iodometry method[47]. Specifically, 1 mL of solution sampled from the reactor was added to 1 mL of 0.4 M potassium iodide (KI) aqueous solution and 1 mL of 0.1 M C$_8$H$_5$KO$_4$ aqueous solution. Then the solution was analyzed by a UV-visible spectrophotometer (Shimadzu UV/Vis 1240, Japan) after 0.5 h in the dark.

### Characterizations

The X-ray diffraction (XRD) patterns of the samples were obtained on an X-ray diffractometer (D/Max-RB, Rigaku, Japan). The Fourier transform infrared (FTIR) spectra and in situ diffuse reflectance infrared Fourier transform spectroscopy (DRIFTS) were recorded on a spectrometer (Nicolet iS50, Thermo Scientific, USA). For the FTIR test of H$_2$O and H$_2$O$_2$, the respective liquid was thoroughly absorbed with potassium bromide (KBr) and analyzed for its FTIR spectrum. Solid state $^{13}$C CP/MAS NMR experiments were conducted using a Bruker Advance 400 MHz spectrometer at a MAS rate of 10 kHz. The thermogravimetric analysis was performed by DTG-60 (SHIMADZU, Japan) over the temperature range from 30 to 600 °C at a heating/cooling rate of 10 °C min$^{-1}$ in air. The BET-specific surface area was measured using an ASAP 2020 nitrogen adsorption apparatus (Micromeritics Instruments, USA). The UV–vis diffuse reflectance spectra (UV–vis DRS) were obtained with a UV–vis spectrophotometer (UV2600, Shimadzu, Japan). The steady-state photoluminescence (PL) was performed on an F-7000 fluorescence spectrophotometer (HITACH, Japan). The time-resolved fluorescence (TRFL) spectra were acquired using an FLS1000 fluorescence lifetime spectrophotometer (Edinburgh, Instruments, UK). The electrochemical measurements were performed on a CHI660C electrochemical workstation (Chenhua, Shanghai), and a three-electrode cell system with a glassy carbon electrode was used as the working electrode, Ag/AgCl electrode as the reference electrode, and platinum foil as the counter electrode. Photocurrent measurement was carried out under light illumination

with light on-off cycles at an open-circuit potential. The photo-irradiated Kelvin probe force microscopy (KPFM) (SPM-9700, Shimadzu, Japan) was used to test the surface potential of the samples. Ultraviolet photoelectron spectroscopy (UPS) and in situ X-ray photoelectron spectroscopy (XPS) were performed on an electron spectrometer (ESCALAB 210, VG, UK). Electron paramagnetic resonance (EPR) was performed on an ESR spectrometer (MEX-nano, Bruker) with a modulation frequency of 100 kHz and a microwave power of 15 mW, 5,5-dimethyl-*l*-pyrroline N-oxide (DMPO) was used to capture superoxide radical (•O$_2^-$) and hydroxyl radical (•OH). All theory calculations have been carried out through the density function theory (DFT) at the level of B3LYP/6-31 G (d, p) within the Gaussian 09 program package. The characteristic molecular fragments of KDBT and KDBT-A were specifically calculated as models to elucidate the D–A property within polymers (Supplementary Tables 4–7 and Supplementary Data 1–4).

### Isotopic analysis

The experiments were performed in 10 mL headspace vials using a 300 W Xenon arc lamp irradiated at the bottom. In a typical photocatalytic reaction, 2 mg of the sample was dispersed in 1 mL of H$_2^{18}$O (90%, Wuhan Liutong Instrument Equipment Co., Ltd.) containing AgNO$_3$ (2 mg) as an electron acceptor and La$_2$O$_3$ (2 mg) as a pH buffer agent[48]. The airtight system was completely evacuated by using a vacuum pump. Then -80 kPa of high-purity N$_2$ gas was injected. The solution was first stirred for 1 h to reach absorption equilibrium and then irradiated with LED (PCX50C, Beijing Perfectlight Technology Co., Ltd, white light, 3 W). After 1 h of irradiation, the gas production was analyzed by gas chromatography-mass spectrometer (GC–MS, 7697 A Headspace Sampler, Agilent).

### Transient absorption measurements

The transient absorption spectroscopy was conducted on a pump-prob system. In detail, the 800 nm output pulse of a 1 kHz Ti: sapphire regenerative amplifier (Coherent) was split into two parts with a 50% beam splitter. One part was transmitted to the Optical Parametric Amplifier (OPA, TOPAS) to generate a pump beam and chopped by a synchronized chopper at 500 Hz. The other part was further split and transformed into white light, serving as a probe beam (less than 10% energy, 420-800 nm) and reference light. Adjusting the time delay between pump and probe beam by a motorized optical delay line, the absorbance change can be calculated with two adjacent probe pulses (pump-blocked and pump-unblocked).

Quantitative samples were dispersed in ultrapure water (or scavenger solutions) with a concentration of 0.2 g L$^{-1}$. The mixture was added into quartz cuvettes with a path length of 2 mm. Cuvettes were sealed using rubber septa caps and degassed with Ar for 10 min. All data were obtained using an excitation wavelength of 400 nm and optical power of 260 μW cm$^{-2}$.

The decay curves obtained from the TA spectra were fitted by the following multi-exponential Eq. (1):

$$I_{(t)} = I_{(0)} + \sum_{i=1}^{n} A_i \exp(-t/\tau_i) \qquad (1)$$

where $I_0$ represents the baseline correction value, and $t$ is the probe time delay. $A_i$ and $\tau_i$ are amplitudes and decay times, respectively. The minimum number of components $n$ to satisfactorily fit the experimental data is three.

## Data availability

Data is available from the authors on request. All data generated in this study are provided in the Source Data file. Source data are provided with this paper.

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

## Acknowledgements

We acknowledge the financial support from the National Key Research and Development Program of China (2022YFB3803600 (J.Y.)) and 2022YFE0115900 (J.Y.); the National Natural Science Foundation of China (52322214 (L.Z.), 22278383 (L.Z.), 22238009 (J.Y.), 22278324 (J.Y.) and 52073223 (J.Y.)); the National Science Foundation of Hubei Province of China (2022CFA001 (J.Y.) and 2023AFA088 (L.Z.)).

## Author contributions

Conceptualization, J.Y., C.C. Investigation, J.Y., C.C., L.Z., M.J., D.X., G.L., and L.W. Writing—Original Draft, J.Y., C.C., L.Z., and M.J. Writing—Review & Editing, J.Y., C.C., L.Z., and M.J. Funding Acquisition, J.Y. and L.Z. Supervision, J.Y.

## Competing interests

The authors declare no competing interests.
