## [Peer Review File · Nature Communications]

In-situ formatting donor-acceptor polymer with giant dipole moment and ultrafast exciton separationREVIEWER COMMENTS

Reviewer #1 (Remarks to the Author):

The manuscript describes the synthesis of a methylene-connected DBT polymer and its in-situ transformation to a D-A semiconducting structure during photocatalytic reactions. The manuscript contains several problems. Followings should be considered.

1) The authors claimed that the electron conductivity of the polymer is increased by the formation of DBT-sulfone units after the photocatalytic reaction. However, the biggest problem is that it is unclear whether the D-A structure created by the photocatalytic reaction is maintained stably during the reactions. The present polymer contains several easily-oxidizable units (S-atom of the DBT units, and the methylene linkers), which may readily be oxidized during the photocatalytic reactions. There is no electrochemical data presented to support the stable oxidation-reduction cycles.

2) Fig. 2c. In the present polymer, DBT and DBT-sulfone units are connected through the un- π -conjugated methylene linkers. This means that entire D-A structure is not created in the present system. As shown in Figure S6, almost no change in absorption spectra after photoirradiation indicates the absence of entire D-A structure.

Based on the above, I do not think the quality of this manuscript justify publication in this journal.

Reviewer #2 (Remarks to the Author):

In this manuscript, the authors present an in-situ study of the transformation from D-D to D-A structure and the enhanced properties of the achieved D-A structure. Interestingly, the authors associate the exciton splitting behavior with a photophysical process of $S_1 \rightarrow S_n$, and fs-TA spectroscopy is used to characterize its transient dynamics. This manuscript is well-written and presents a significant contribution to the field. The research is thorough and well-documented, and the results are clearly presented. The methodology appears robust, and the data is convincing. Thus, I recommend acceptance of this paper to be published in Nature Communications with minor revision. I only have a few suggestions for further improvement.

1. It is crucial to clearly characterize the molecular structure for this work. In addition to FTIR and XPS, more measurements should be presented to verify the successful synthesis of KDBT and KDBT-A, such as ^{13}C cross-polarization/magic angle spinning nuclear magnetic resonance (CP/MAS NMR).
2. While the transformation from D-D to D-A is interesting, it would be helpful to clarify the advantages of such a transformation for photocatalysis. Can the D-A structure be synthesized directly, and if so, what are the benefits of the transformation process?
3. How many DBT units were oxidized to DBTSO? Can it be quantified?
4. The HOMO and LUMO levels of the polymers could be better measured using cyclic voltammetry.
5. Is it possible to switch the obtained D-A polymer back to a D-D structure after irradiation?

6. While photocatalytic H₂O₂ production is not the main focus of this work, providing some insights into the photocatalytic mechanism and a thorough literature survey with a comparison table of published results would enhance the manuscript.
7. What is the physic meaning of contact potential differences (CPDs)? Why the CPDs value was increased under UV light irradiation?
8. The synthetic details should be more specific, including information on temperature, ultrasonic power, and other relevant parameters.

Reviewer #3 (Remarks to the Author):

In this manuscript, the authors have developed a post-synthetic method that in situ convert donor-carbon-donor (KDBT) into donor-carbon-acceptor (KDBT-A) polymer under visible light. The resulting polymer exhibits a large intramolecular dipole moment and fast exciton separation. The superior photocatalytic performance of KDBT-A is demonstrated in the generation of hydrogen peroxide. Finally, the mechanistic experiments have been thoroughly investigated. Although this is a nice work, some issues should be addressed.

1) Page 11, line 166:

In the text "It shows that a part of dibenzothiophene units is oxidized to...", What's the accurate percentage of oxidation? Based on the structure-activity relationship, the photocatalytic performance will vary depending on the degree of oxidation.

2) Page 10, Figure 2b:

Apart from the first cycle, comparable efficiency was achieved for the production of hydrogen peroxide in the following cycles. Does it mean the polymeric photocatalyst will not change after the first run? In theory, the ratio for oxidation of S will increase as the reaction proceeds, it is recommended to monitor the ratio of SO₂ in the polymeric photocatalyst after each cycle.

3) In the characterization part, solid-state ¹³C CP/MAS NMR spectroscopy of KDBT can be added.

4) Fig 1b-1f: More information should be given, such as how to prepare the sample for TEM, how to form the nanotube?

5) The structures drawn in Fig 1a and Fig 2c may be misunderstood as linear polymers, it is better to clearly show the hyper-cross-linked structures.

Response to reviewer's comments

Reviewer #1

The manuscript describes the synthesis of a methylene-connected DBT polymer and its in-situ transformation to a D-A semiconducting structure during photocatalytic reactions. The manuscript contains several problems. Followings should be considered.

1) The authors claimed that the electron conductivity of the polymer is increased by the formation of DBT-sulfone units after the photocatalytic reaction. However, the biggest problem is that it is unclear whether the D-A structure created by the photocatalytic reaction is maintained stably during the reactions. The present polymer contains several easily-oxidizable units (S-atom of the DBT units, and the methylene linkers), which may readily be oxidized during the photocatalytic reactions. There is no electrochemical data presented to support the stable oxidation-reduction cycles.

Response: In response to the reviewer's feedback, we conducted XPS and ^{13}C CP/MAS NMR tests after each cycle. As depicted in Figure R1 (same as Supplementary Figure 14), the sulfur atoms underwent oxidation to form SO_2 units, while the sulfone units demonstrated robustness in subsequent cycles, as evidenced by the percentage increase in the newly emerged S 2p XPS peak. Additionally, Figure R2 (same as Supplementary Figure 4) presents a comparison of the ^{13}C CP/MAS NMR spectra of the polymer before and after each cycle. In all spectra, prominent resonance peaks at approximately 138 and 124 ppm are indicative of substituted and unsubstituted aromatic carbons, respectively. The peak at around 39 ppm corresponds to the methylene linker carbons. Notably, the signal from the methylene linker remains consistent after the cycling tests, suggesting the structural stability of the molecular skeleton for photocatalytic processes

Figure R1. S 2p XPS spectra of the polymer during the cycle tests. The percentage of the sulphonyl group S 2p signal remained within the range of 25% to 30%.

Figure R2. ^{13}C CP/MAS NMR spectra of the polymer during the cycle tests. Asterisks denote spinning sidebands.

2) Fig. 2c. In the present polymer, DBT and DBT-sulfone units are connected through the un- π -conjugated methylene linkers. This means that entire D-A structure is not created in the present system. As shown in Figure S6, almost no change in absorption spectra after photoirradiation indicates the absence of entire D-A structure.

Response: The D-A strategy is flourishing within the conjugated system, where the dipole moment between the donor and acceptor units is the key factor for the formation of the D-A structure. A large dipole moment can trigger push-pull intramolecular charge transfer. Recently, resorcinol–formaldehyde (RF) resins, where the donor (resorcinol) and acceptor (formaldehyde) units are also connected through methylene linkers, were presented as effective photocatalysts for hydrogen peroxide production (Nat. Mater. 2019, 18, 985–993). Further, Zhu and coworkers introduced a σ -linkage into the fully π -conjugated D-A polymer, thus controlling π - π stacking distance and enhancing the photocatalytic performance (Angew. Chem. Int. Ed. 2023, 62, e202304773). Hence, theoretically, the D-A structure can be created by our method. Despite the absence of observable changes in the absorption spectra between KDBT and KDBT-A, significant differences were evident in other spectra, including XPS and PL. Moreover, we conducted in situ diffuse-reflectance infrared Fourier transform spectroscopy (DRIFTS) to monitor the irradiation process. As depicted in **Figure R3 (same as Supplementary Figure 13)**, the negative signal (depicted in blue) around $\sim 3625\text{ cm}^{-1}$ indicates the depletion of surface H_2O , while the emergence of a positive signal (depicted in orange-red) at $\sim 3250\text{ cm}^{-1}$ corresponds to the generation of H_2O_2 . Notably, the emergence of a signal around $\sim 1200\text{ cm}^{-1}$ signifies the gradual formation of the sulphonyl group ($\text{O}=\text{S}=\text{O}$), providing compelling evidence for the creation of the D-A structure.

Figure R3. In-situ DRIFTS spectra of KDBT. The negative signal (blue color) represents the consumption of species, and the positive signal (orange-red color) represents the generation of new species.

Reviewer #2

In this manuscript, the authors present an in-situ study of the transformation from D-D to D-A structure and the enhanced properties of the achieved D-A structure. Interestingly, the authors associate the exciton splitting behavior with a photophysical process of $S_1 \rightarrow S_n$, and fs-TA spectroscopy is used to characterize its transient dynamics. This manuscript is well-written and presents a significant contribution to the field. The research is thorough and well-documented, and the results are clearly presented. The methodology appears robust, and the data is convincing. Thus, I recommend acceptance of this paper to be published in Nature Communications with minor revision. I only have a few suggestions for further improvement.

1. It is crucial to clearly characterize the molecular structure for this work. In addition to FTIR and XPS, more measurements should be presented to verify the successful synthesis of KDBT and KDBT-A, such as ^{13}C cross-polarization/magic angle spinning nuclear magnetic resonance (CP/MAS NMR).

Response: According to the reviewer's suggestion, we have performed the ^{13}C CP/MAS NMR tests on KDBT and KDBT-A after each cycle. As shown in Figure R4 (same as Supplementary Figure 4), all the samples show strong resonance peaks at ~ 138 and ~ 124 ppm that can be ascribed to the substituted aromatic C and unsubstituted aromatic C. The

peak at ~39 ppm is interpreted as the carbon of methylene linkers. The signals marked with asterisks are spinning sidebands.

Figure R4. ^{13}C CP/MAS NMR of the polymer during the cycle tests. Asterisks denote spinning sidebands.

2. While the transformation from D-D to D-A is interesting, it would be helpful to clarify the advantages of such a transformation for photocatalysis. Can the D-A structure be synthesized directly, and if so, what are the benefits of the transformation process?

Response: Previously, the synthesis of most D-A (donor-acceptor) polymers involved direct methods, demonstrating superior performance compared to corresponding D-D (donor-donor) polymers in fields such as photocatalysis, solar cells, and OLEDs. These studies often attempted to correlate performance trends with spectral absorption and HOMO-LUMO positions. Despite these achievements, several unresolved issues persist. Firstly, direct bottom-up synthesis is intricate and costly, limiting its scalability for practical applications. Secondly, attributing superior spectral absorption and higher HOMO-LUMO positions in D-A polymers solely to the electronic push-pull (D-A) effect is challenging due to the inability to ensure consistent polymerization degrees across batches synthesized separately. Conversely, our approach introduces a sustainable, light-induced post-synthetic method for crafting the D-A structure (KDBT-A). This method is both simple and cost-effective. Crucially, it enables the in-situ formation of the D-A structure while preserving the polymer backbone (degree of polymerization) of the as-synthesized polymer.

3. How many DBT units were oxidized to DBTSO? Can it be quantified?

Response: The oxidized DBT units can be roughly quantified in XPS spectra. The percentage of newly emerged S 2p peak over KDBT-A, which reflected the formation of DBTSO, is calculated to be 25-30%.

4. The HOMO and LUMO levels of the polymers could be better measured using cyclic voltammetry.

Response: According to the reviewer's suggestion, we have measured the CV curves of KDBT and KDBT-A. As shown in Figure R5 (same as Supplementary Figure 9), the ferrocene (Fc) was used as an internal standard, which was assigned an absolute energy of -4.8 eV vs. vacuum level. The HOMO levels (vs. vacuum) were calculated by the equation (eq. R1).

$$E_{HOMO} = -e \times (E_{ox} + 4.8 - E_{ox}^{Fc/Fc^+}) \quad (R1)$$

where E_{ox} and E_{ox}^{Fc/Fc^+} are the onset oxidation potential of polymers and ferrocene vs. Ag/AgCl.

Figure R5. The CV curves of ferrocene, KDBT, and KDBT-A.

5. Is it possible to switch the obtained D-A polymer back to a D-D structure after irradiation?

Response: In this study, the transition from the D-D structure (KDBT) to the D-A structure was achieved efficiently. Further irradiation did not induce changes in the D-A polymer (as confirmed by XPS tests). Our experiments demonstrated that the monomer DBT readily oxidizes to DBTSO in the presence of H_2O_2 . Conversely, the reduction of monomer DBTSO back to DBT proved challenging (requiring $LiAlH_4$ as a reducing agent). Consequently, it can be inferred that restoring KDBT-A to KDBT would be even more challenging.

6. While photocatalytic H_2O_2 production is not the main focus of this work, providing some insights into the photocatalytic mechanism and a thorough literature survey with a

comparison table of published results would enhance the manuscript.

Response: In response to the reviewer's suggestion, we conducted in situ diffuse-reflectance infrared Fourier transform spectroscopy (DRIFTS) to delve deeper into the photocatalytic mechanism (Figure R6, same as Supplementary Figure 13). The negative signal (depicted in blue) around 3625 cm^{-1} indicates the consumption of surface H_2O , while the emergence of a positive signal (depicted in orange-red) at $\sim 3250\text{ cm}^{-1}$ correlates with the generation of H_2O_2 . Remarkably, we observed a gradual increase in a band around $\sim 1200\text{ cm}^{-1}$, which corresponds to the characteristic signal of the sulphonyl group ($\text{O}=\text{S}=\text{O}$). These findings offer robust evidence of structural transformation. The performance evaluation of photocatalytic H_2O_2 production on polymer catalysts is outlined in Table R1 (same as Supplementary Table 2).

Figure R6. In-situ DRIFTS spectra of KDBT. The negative signal (blue color) represents the consumption of species, and the positive signal (orange-red color) represents the generation of new species.

Table R1. A comparison of photocatalytic H_2O_2 production activity of the present work with other polymer photocatalysts.

photocatalyst	Condition	Rate ($\mu\text{M h}^{-1}$) ^a	Ref.
CHF-DPDA	O_2	3450	4
TTF-BT-COF	O_2	1380	5
N_0 -COF	O_2	785	6
RF523	O_2	138	7

Bpy-TAPT	O ₂	686	8
CTF-BDDBN	O ₂	58	9
HEP-TAPT-COF	O ₂	875	10
MRF-250	O ₂	972	11
6DEPI	O ₂	97	12
TDB-COF	O ₂	724	13
KDBT	Air	870	This work
KDBT-A	Air	1320	This work

^a The unit of H₂O₂ production rate is standardized to μM h⁻¹.

7. What is the physical meaning of contact potential differences (CPDs)? Why the CPDs value was increased under UV light irradiation?

Response: When two different materials are brought into electrical contact, their Fermi level will coincide, thus resulting in a potential difference called the contact potential differences (CPDs). In this case, the CPD stems from the different work functions between the conducting Kelvin tip (W_t) and sample surface (W_s , eq. R2).

$$CPD = \frac{W_t - W_s}{e} \quad (R2)$$

Upon light irradiation, the work function of the Kelvin tip is constant, while the sample is excited, namely, the work function of the sample is decreased. Hence, the CPD value was increased under UV light irradiation.

8. The synthetic details should be more specific, including information on temperature, ultrasonic power, and other relevant parameters.

Response: According to the suggestion, more synthetic details are presented.

Reviewer #3

In this manuscript, the authors have developed a post-synthetic method that in situ convert donor-carbon-donor (KDBT) into donor-carbon-acceptor (KDBT-A) polymer under visible light. The resulting polymer exhibits a large intramolecular dipole moment and fast exciton separation. The superior photocatalytic performance of KDBT-A is demonstrated in the generation of hydrogen peroxide. Finally, the mechanistic experiments have been thoroughly investigated. Although this is a nice work, some issues should be addressed.

1) Page 11, line 166:

In the text "It shows that a part of dibenzothiophene units is oxidized to...", What's the accurate percentage of oxidation? Based on the structure-activity relationship, the photocatalytic performance will vary depending on the degree of oxidation.

Response: According to the reviewer's suggestion, we have calculated the percentage of newly emerged S 2p peak in XPS spectra, which reflects the degree of oxidation. As a result, the progression of oxidation percentages was noted as 0-6%-9%-15%-26%.

2) Page 10, Figure 2b:

Apart from the first cycle, comparable efficiency was achieved for the production of

hydrogen peroxide in the following cycles. Does it mean the polymeric photocatalyst will not change after the first run? In theory, the ratio for oxidation of S will increase as the reaction proceeds, it is recommended to monitor the ratio of SO₂ in the polymeric photocatalyst after each cycle.

Response: Yes, the polymeric photocatalyst is stable after the initial cycle. We have performed the XPS test and calculated the ratio of the sulphonyl group after each cycle. As shown in **Figure R7** (same as Supplementary Figure 14). The ratio of SO₂ is consistently maintained between 25-30% across the subsequent cycles.

Figure R7. S 2p XPS spectra of the polymer during the cycle tests. The percentage of the sulphonyl group S 2p signal remained within the range of 25% to 30%.

3) In the characterization part, solid-state ¹³C CP/MAS NMR spectroscopy of KDBT can be added.

Response: According to the reviewer's suggestion, we have performed the ¹³C CP/MAS NMR tests on KDBT and KDBT-A after each cycle. **Figure R8** (same as Supplementary Figure 4) presents a comparison of the ¹³C CP/MAS spectra of the polymer before and after each cycle. In all spectra, prominent resonance peaks at approximately 138 and 124 ppm are indicative of substituted and unsubstituted aromatic carbons, respectively. The peak at around 39 ppm corresponds to the methylene linker carbons. The signals marked with asterisks are spinning sidebands.

Figure R8. ^{13}C CP/MAS NMR of the polymer during the cycle tests. Asterisks denote spinning sidebands.

4) Fig 1b-1f: More information should be given, such as how to prepare the sample for TEM, how to form the nanotube?

Response: The experimental procedures strongly impact the samples' morphology. We conducted a series of simple methodological experiments to investigate their influence (Table R2 and Figure R9). Table R2 is added to supplementary materials as Supplementary Table 3.

Table R2. Effect of monomer concentration and duration of ultrasonication on the morphology of KDBT polymer.

Conc. ^a \ Time ^b	2	3	4
0.02	particle	particle/tube	sphere/tube
0.025	sphere/tube	tube	tube
0.05	sphere	sphere/tube	bulk

^a Concentration of DBT units (mol/L).

^b Duration of Ultrasonication (h).

Figure R9. SEM images of KDBT polymer under different synthetic conditions: (a) 0.02 M DBT for 2 h, (b) 0.02 M DBT for 3 h, (c) 0.02 M DBT for 4 h, (d) 0.025 M DBT for 2 h, (e) 0.025 M DBT for 3 h, (f) 0.025 M DBT for 4 h, (g) 0.05 M DBT for 2 h, (h) 0.05 M DBT for 3 h and (i) 0.05 M DBT for 4 h.

5) The structures drawn in Fig 1a and Fig 2c may be misunderstood as linear polymers, it is better to clearly show the hyper-cross-linked structures.

Response: Thank you for the helpful suggestions. Typically, aromatic compounds like benzene, thiophene, pyrrole, and furan can readily undergo crosslinking to create hyper-cross-linked porous structures owing to their high electrophilic aromatic substitution activity. However, the DBT unit differs from the typical aromatic ring. Protonation (electrophilic attack) of DBT predominantly occurs at the 2 (8)-positions (ChemPhysChem 2011, 12, 2870–2885), which leads to a tendency for linear propagation in the polymer.

REVIEWER COMMENTS

Reviewer #1 (Remarks to the Author):

The problems raised by the first round of review were not addressed by the authors. Followings should be considered.

1) The question that must be clarified is whether the polymer stably catalyzes water oxidation and O₂ reduction, as denoted by the authors in Figure S17. Based on the structure of the polymer, the most oxidizable part must be an exposed -S- atom of the DBT unit or a methylene unit of the polymer. In addition, the valence band level (~1.8 V) is insufficient for water oxidation. Furthermore, there is no evidence for O₂ evolution on the catalyst. In addition, the anodic current observed in Figure S9 may originate from the -S- oxidation. The authors must provide the evidence for water oxidation by the holes by a half-photoreaction using a sacrificial electron acceptor. The authors carefully reconsider a correct mechanism for photocatalysis on the presented polymer.

2) The newly provided Figure S13 involves incorrect assignment. The decreased 3400-3700 cm⁻¹ peak is assigned to the H-bonding -OH. The increased 3000-3500 cm⁻¹ peak is assigned to bulk water, which may be formed by the reduction of O₂ molecule. H₂O₂ does not show a peak at 3000-3500 cm⁻¹. Careful assignment is necessary.

At the present form, it is unclear whether the polymer stably promote water oxidation and O₂ reduction. I think the most plausible conclusion is that this polymer simply promotes O₂ reduction through self-oxidation. Many photocatalyst researchers will probably have similar impressions. I do not think the quality of this manuscript justify publication in this journal.

Reviewer #2 (Remarks to the Author):

The authous have addressed properly the issues that I raised, and the quality of manuscript has been improved,it could be accepted as it is.

Reviewer #3 (Remarks to the Author):

The authors have addressed the issues, therefore the revised manuscript is suggested be be accepted for publication.

Reviewer #1 (Remarks to the Author):

The problems raised by the first round of review were not addressed by the authors. Followings should be considered.

1) The question that must be clarified is whether the polymer stably catalyzes water oxidation and O₂ reduction, as denoted by the authors in Figure S17. Based on the structure of the polymer, the most oxidizable part must be an exposed -S- atom of the DBT unit or a methylene unit of the polymer. In addition, the valence band level (~1.8 V) is insufficient for water oxidation. Furthermore, there is no evidence for O₂ evolution on the catalyst. In addition, the anodic current observed in Figure S9 may originate from the -S-oxidation. The authors must provide the evidence for water oxidation by the holes by a half-photoreaction using a sacrificial electron acceptor. The authors carefully reconsider a correct mechanism for photocatalysis on the presented polymer.

Response: The reviewer's concerns about the stability of the polymers have already been addressed in the first round of review. In response, we conducted additional tests, including ¹³C CP/MAS NMR spectra analysis, which verified the stability of methylene under continuous irradiation. Regarding the sulfur atom, our findings confirm that the pristine polymer (KDBT) undergoes self-oxidation at the sulfur site early in the irradiation process and transforms to KDBT-A. Subsequent photocatalysis over KDBT-A exhibited a steady H₂O₂ production, demonstrating the stability of the exposed sulfur atom, as evidenced by the XPS test.

Regarding the concern about the valence band level's adequacy for water oxidation, we respectfully disagree with the assessment that the valence band level (~1.8 V) is insufficient. Theoretically, the redox potential for water-to-oxygen conversion is 1.23 V versus NHE (pH = 0, eq. R1). Hence, the HOMO levels of the polymers (~1.87 V) fulfill the thermodynamic requirement of water oxidation. Numerous studies in the literature have also shown that polymers with even higher HOMO positions are capable of producing oxygen from water, underscoring the potential of our samples for photocatalytic oxygen production. For instance, Zhang and coworkers (Nat. Commun. 2019, 10, 2467) developed a g-C₄₀N₃-COF photocatalyst and tested it for water oxidation in the presence of silver nitrate (AgNO₃) as an electron acceptor. As shown in Figure R2, despite the VB potential of the material being less than +1.8 V versus NHE (pH = 0, standardized by the Nernst equation), the bare g-C₄₀N₃-COF exhibits significant O₂ generation.

Our experimental and theoretical results, in conjunction with analogous findings from Lan et al. (Nat. Commun. 2023, 14, 593), substantiate the validity of our proposed mechanism. Specifically, Lan and colleagues reported three COFs for photocatalytic overall water splitting, exhibiting mechanisms similar to those described in our manuscript. As illustrated in Figure R1, except for the TpBD-NS sample with the smallest HOMO potential, the other COFs continuously catalyzed O₂ evolution.

Figure R1. (a) The band structures and (b) O₂ evolution of the COFs. (Ref. Nat. Commun. 2023, 14, 593)

Figure R2. (a) The band structures and (b) O₂ evolution of the material. (Ref. Nat. Commun. 2019, 10, 2467)

The production of O₂ has already been verified through isotope analysis (refer to Supplementary Figs. 16 and 17) in the original manuscript. To address the reviewer's suggestions, we further conducted half-photoreaction in a Pyrex top-irradiation reaction vessel connected to a glass-closed gas circulation system. In detail, 20 mg of KDBT-A was dispersed in 10 mL deionized water containing AgNO₃ (0.01 M) as an electron acceptor and 20 mg La₂O₃ as a pH buffer agent. The reactant solution was evacuated several times to completely remove air prior to irradiation under a 300 W Xe lamp and a water-cooling filter. The evolved gases were analyzed by gas chromatography equipped with a Barrier Discharge Ionization Detector (BID). The resultant **Figure R3** demonstrates the continuous production of O₂ on the catalyst. These findings, when combined with our earlier results (isotope analysis, XPS, and NMR spectra), strongly support our conclusion that the KDBT-A polymer will oxidize H₂O into O₂ rather than self-oxidizing.

Figure R3. Photocatalytic O₂ evolution experiments.

2) The newly provided Figure S13 involves incorrect assignment. The decreased 3400-3700 cm⁻¹ peak is assigned to the H-bonding -OH. The increased 3000-3500 cm⁻¹ peak is assigned to bulk water, which may be formed by the reduction of O₂ molecule. H₂O₂ does not show a peak at 3000-3500 cm⁻¹. Careful assignment is necessary.

Response: In response to the reviewer's concerns about the assignment of signals in Figure S13, we took steps to refine the signal assignments for clarity and accuracy. To achieve this, we conducted IR spectra measurements on both H₂O and H₂O₂. In each case, the respective liquid was thoroughly absorbed using potassium bromide (KBr) and analyzed for its infrared spectrum. As shown in Figure R4a, both H₂O and H₂O₂ display strong absorption signals beyond 3000 cm⁻¹. To ensure a more precise assignment, we directed our focus towards the characteristic peak of H₂O₂, specifically identified at ~2850 and 1380 cm⁻¹. Figures R4b and c present the distinct emergence of signals at ~2850 and 1380 cm⁻¹, aligning with the gradual formation of H₂O₂. Given these findings, Supplementary Figure 13 has been replaced by Figure R4.

Figure R3. (a) FTIR spectra depicting comparisons between H₂O and H₂O₂. (b) In-situ DRIFTS spectra showcasing signals within the 2500-4000 cm⁻¹ range. (c) In-situ DRIFTS spectra illustrating signals spanning the 900-1500 cm⁻¹ range.

REVIEWERS' COMMENTS

Reviewer #1 (Remarks to the Author):

The problems were addressed by the authors. Now I recommend the manuscript for publication.